# Novel Antibacterial Metals as Food Contact Materials: A Review

**DOI:** 10.3390/ma16083029

**Published:** 2023-04-11

**Authors:** Xinrui Zhang, Chunguang Yang, Ke Yang

**Affiliations:** Institute of Metal Research, Chinese Academy of Sciences, Shenyang 110016, China

**Keywords:** food contact material, bacterial contamination, antibacterial metals and alloys, antibacterial stainless steel

## Abstract

Food contamination caused by microorganisms is a significant issue in the food field that not only affects the shelf life of food, but also threatens human health, causing huge economic losses. Considering that the materials in direct or indirect contact with food are important carriers and vectors of microorganisms, the development of antibacterial food contact materials is an important coping strategy. However, different antibacterial agents, manufacturing methods, and material characteristics have brought great challenges to the antibacterial effectiveness, durability, and component migration associated with the use security of materials. Therefore, this review focused on the most widely used metal-type food contact materials and comprehensively presents the research progress regarding antibacterial food contact materials, hoping to provide references for exploring novel antibacterial food contact materials.

## 1. Introduction

Food contact materials play a vital role in food supply chains [1]. Appropriate materials can be a barrier to protect food from collision, friction, and other damage in order to keep the moisture, composition, quality, and other characteristics of the food itself unchanged, even protecting the food from external pollution [2,3]. At present, the food contact materials allowed in China mainly include plastic, rubber, enamel, metal, glass, packaging paper, and composite films [4]. Among these, metal food contact materials and products have the widest and the largest amount of use. Additionally, according to the material, metal food contact materials can be divided into stainless steel, aluminum, iron, copper, gold, and silver. Stainless steel and aluminum products are the most widely used, covering tableware, kitchenware, food packaging, food production equipment, and so on. Generally speaking, the function of food contact materials is relatively simple, and they are usually biologically inert. Though their usage can isolate the food from the surrounding environment to a certain extent, they cannot effectively keep the food fresh. What is more, microorganisms, including various spoilage and foodborne pathogenic bacteria, can attach to solid surfaces, such as plastic, glass, metal, and wood, during food processing, transportation, and storage [5]. Once bacteria form biofilms, their sensitivity to chemical fungicides and environmental changes is significantly reduced, and their heat resistance is correspondingly increased, allowing them to easily escape disinfection [6]. This means that food contact materials can easily become carriers and media for pathogenic bacteria and spoilage bacteria, affecting the shelf life and even the safety of food, eventually leading to serious health problems and economic losses [7].

With higher requirements of consumers for food quality and safety, and the development of material science and technology, active materials have emerged and attracted great attention in the food field [8]: for example, active packaging, which incorporates active substances into existing food packaging materials. When the modified materials are used for packing, they can absorb water, oxygen, and other elements unfavorable to freshness from the food storage environment through the actions of the active substances, extending the shelf life of food [9,10]. Furthermore, some of them can actively adjust the interaction between the food and the surrounding environment for the purpose of protecting the food quality [11,12]. Nowadays, the research on active food contact materials mainly focuses on the atmospheric regulation of fruit and vegetable packaging for maintaining freshness, such as packaging materials with oxygen absorption/release, carbon dioxide absorption/release, ethylene removal, humidity control/anti-fog, and anti-oxidation abilities [13].

In fact, although there are many factors that accelerate food spoilage, microorganisms are the main cause [14]. Microorganisms are ubiquitous in food supplies. Specifically, there are a relatively high number of initial bacteria in the raw materials of food, and food processing environments are rich in nutrients, which promotes the proliferation of microorganisms. In particular, some surfaces, such as internal equipment, complex parts, and pipes, are not easy to clean, which makes it easy for bacteria to accumulate and form biofilms [15]. A biofilm is a special cell population structure formed by microorganisms colonized on a solid surface that gradually wrap themselves in their own secreted extracellular polymeric substances (EPSs). EPSs provide shelter and nutrients for bacteria in the biofilm, greatly increasing the resistance of these bacteria to the external environment [16,17]. Studies have shown that biofilm formation in food environments such as floor drains, storage tanks, wheelbarrows, and conveyor belts is the main cause of food contamination [18,19]. Accordingly, many cleaning and elimination methods have been developed for food processing environments. For example, a cleaning-in-place (CIP) program has been adopted in dairy processing lines. In the CIP program, mechanical brushes, chemical detergents, and even biological enzymes are used to assist the cleaning in order to obtain a biofilm-free environment as far as possible. However, even with efficient cleaning systems such as CIP treatment, microbial residues and growth into biofilms cannot be completely avoided [20]. Gibson et al. tested biofilm resistance to disinfectants and found that commercially available Easyclean (an alkaline disinfectant) and Ambersan (an acidic disinfectant) were not effective in removing *Pseudomonas aeruginosa* and *Staphylococcus aureus* (*S. aureus*) biofilms from stainless-steel surfaces and could only reduce the biomass of the biofilm by an order of magnitude [21]. Even worse, various microorganisms are highly adaptable to extreme environments with a supply of food nutrition. For example, microorganisms can invade food stored at a low temperature. *Pseudomonas* can grow in large quantities under cold storage conditions and produce extracellular proteolytic enzymes, causing the spoilage of high-protein foods such as sterilized milk and fish [22]. Moreover, *Pseudomonas* also widely grow in food processing plants, including on cutting boards, in water supply pipeline systems, and in the ground, forming biofilms and resulting in the post-pollution of products during storage [23].

Based on this, the development and application of antibacterial materials for food have attracted great attention. Antibacterial food contact materials are usually composed of antibacterial agents and conventional basic materials. Such materials can inhibit the propagation of microorganisms in food through direct contact with food or the diffusion/migration of antibacterial agents and maintain their own cleanness [24] in order to meet the consumer’s pursuit of high-quality products and the need to prevent various foodborne diseases. The usage of these materials is the most plausible food preservation strategy at present. However, the development of antibacterial food contact materials is still in the initial stages, because the component migration from food contact materials is related to food safety. Therefore, the selection of antibacterial agents is crucial.

Natural antibacterial agents mainly come from animals, plants, or microorganisms, and so pose little harm to human health and the ecological environment. Therefore, they are popular antibacterial agents for active food materials [25,26]. For example, essential oils are secondary metabolites of plants and natural extracts. When they are added to food contact materials by physical adsorption or coating, they can act on the food through volatilization [27]. The antibacterial mechanism of essential oils lies in their hydrophobicity, which promotes their interaction with the bacterial membrane and mitochondria and then destroys the microbial structure, increases the permeability of the cell membrane, and effectively inactivates the microorganism [28]. However, the poor water solubility, volatility, and easily emitted pungent odor limit their application. Lysozyme, as a representative enzymic antibacterial agent, can be extracted from animals, plants, and microorganisms. It can decompose the insoluble mucopolysaccharides of bacterial cell walls into soluble glycopeptides, resulting in cell wall rupture and finally bacterial death. It is known as a safe natural preservative and has great application potential in food preservation [29,30]. However, the practical application value of lysozyme is also limited by its instability and easy deactivation. Chitosan is the product of the deacetylation of chitin produced in crustacean shells or the cell walls of fungi and insects. Chitosan, with its strong inhibitory effect on most pathogenic bacteria, is a nontoxic, biodegradable, and biocompatible cationic polysaccharide that has attracted widespread attention in the field of food and biomedicine [31]. In order to promote the development of chitosan in antibacterial materials for food, researchers have developed a variety of methods for preparing chitosan films [32]. Similar to most natural antibacterial agents, chitosan has poor water solubility in alkaline or neutral conditions, which limits its application in the preparation of antibacterial packaging materials. In short, natural antibacterial agents are usually safe and nontoxic, but their stability and effectiveness need to be improved. By contrast, inorganic antibacterial agents such as Ag, Cu, Zn, and their oxides have become more and more popular in the food field, since they exhibit broad-spectrum antibacterial activity toward nearly all bacteria, and no drug resistance has been reported so far [33]. In fact, inorganic antibacterial agents were used for food contact materials in ancient times. Silver cups, silver chopsticks, and other silver articles have famously been used for poison tests and sterilization. At present, these elements have also been proposed as food contact materials; in particular, some metal-related nanoparticles have been used to modify food packaging [34]. However, the usage of metal food contact materials involves safety issues owing to the migration of toxic and harmful heavy-metal elements, which is the biggest limiting factor for their practical application and development. Therefore, as researchers in the field of metal materials, we comprehensively reviewed and summarized the research into antibacterial food contact materials in this article, hoping to provide more guidance on the development of new generations of functional metal food contact materials. Typical antibacterial agents containing metals or their oxides, such as Ag [32], ZnO [35], titanium dioxide nanoparticles (TiO_2_) [36], and Cu oxide [37], have been made into nano-antibacterial materials for food contact, antibacterial coatings, and alloy products.

## 2. Nano-Antibacterial Materials for Food Contact

Food contact nano-antibacterial materials can combine the basic functions of nanomaterials, such as the surface effect and small size effect, and the multiple advantages of antibacterial materials, including their broad-spectrum antibacterial effect, good heat resistance, and prevention of drug resistance [38]. The specific surface area of nanoparticles is usually large, but the particle size is small, and the bond state ratio is unbalanced. Furthermore, there are many active centers, which enhances the adsorption ability of nanoparticles for oxygen atoms, oxygen free radicals, and other hydrocarbon molecules that promote food degradation and spoilage, ensuring the good antibacterial and antiseptic properties of nanoparticles [39]. In the food field, silver nanoparticles (AgNPs) are one of the mostly commonly used antibacterial agents. AgNPs can adhere to or penetrate into microbial cell membranes, affecting the cell membrane permeability and damaging the bacterial respiratory metabolism [40,41]. Wang et al. found that an antibacterial film with 4% AgNP addition had an antibacterial rate of more than 91.2% compared with an ordinary film, showing a significant preservative effect on cabbage, which manifested in reduced weight loss and leaf yellowing [29]. The polymer-based AgNP composite not only had excellent antibacterial ability, but also retained the properties of the polymer, i.e., malleability and suitable mechanical strength [42,43]. At present, the polymer-based AgNP composite materials used in the market as food contact materials include lunch boxes, boxes for maintaining freshness, milk bottles, and beverage bottles [44]. ZnO NPs are certified as safe nano-antibacterial agents by the US Food and Drug Administration (FDA), and they can also provide essential micronutrients for human health [45,46]. Al-Tayyar et al. reported the use of a biological nanocomposite film prepared by adding a ZnO-SiO_2_ nanocomposite to a poly (vinyl alcohol) (PVA)/chitosan (CS) polymer for food packaging, as shown in Figure 1 [47]. The addition of the ZnO-SiO_2_ nanocomposite lent the polymer film appropriate mechanical properties and good permeability for oxygen and water, which was beneficial for preventing mildew and maintaining the freshness of bread. TiO_2_ is a kind of low-cost metal oxide with high chemical stability. The reactive oxygen species (ROS) produced by TiO_2_ under photocatalysis can cause oxidative damage to microbial cells, resulting in their death, and so this material has been widely used in catalysis and the food and medical fields [48,49]. Zeng et al. prepared a nano-TiO_2_ antibacterial composite preservative film. During the preservation test using loquats, it was found that 2% was the optimal amount of TiO_2_ in the film, providing good film-forming properties, mechanical properties, and antibacterial activity. Even under freezing condition, this material significantly reduced the total bacterial count, decay rate, and oxide content in the loquats, ultimately ensuring good freshness and lengthening the shelf lives of the fruit [50].

## 3. Antibacterial Coating for Food Contact

The coating process is relatively simple. Thus, food contact coatings have been widely used to endow packaging with preservation and antibacterial abilities. The application of antibacterial agents in the food coating industry is an effective way to form an antibacterial surface layer on the coating material, which is an important strategy to endow the material with broad-spectrum antibacterial activity and even improve the quality of the coatings. For example, a coating with nano-TiO_2_ addition can present disinfection abilities under the conditions of photocatalysis, which has been used in food contact materials such as the inner walls of cans and water pipes. The related coating technology is relatively mature. In recent years, there have been commercial applications of antibacterial coatings in food packaging/contact materials, such as SongSing’s application of composite coatings containing TiO_2_ and Ag in drinking kettles. Sam Sung applied a nano-Ag coating to the inner wall of a refrigerator, and Baby Dream used a coating containing nano-Ag in baby cups [51,52,53]. Corning developed a Cu-containing glass–ceramic coating that has been demonstrated to exhibit rapid bactericidal and antiviral properties, as shown in Figure 2 [54]. A common disadvantage is that the coatings flake easily, causing a loss in antibacterial properties.

## 4. Antibacterial Alloys for Food Contact

An antibacterial alloy is a kind of metallic material that exhibits a strong ability to inhibit the adhesion, growth, and proliferation of bacteria through proper element alloying, metal forming, and heat treatment [55]. Recently, Cu- and Ag-containing antibacterial metal alloys have been reported to exhibit a strong antibacterial ability against many bacteria, such as antibacterial stainless steel [56], antibacterial titanium alloy [57,58], antibacterial magnesium alloy, and antibacterial cobalt alloy [59,60,61]. The main alloying elements are Ag and Cu, which are well-known to have strong and broad-spectrum antibacterial abilities. In comparison with antibacterial coatings or nanoparticle-containing materials, antibacterial alloys possess the following advantages:Long-term antibacterial ability

Since the whole alloy exhibits antibacterial activity, the antibacterial ability is not lost during machining, wear or abrasion, corrosion, etc. Taking antibacterial stainless steel as an example, the manufacturing process involves adding an appropriate amount of antibacterial Cu or Ag into the ordinary stainless steel during the metallurgical process. Then, the material is subjected to solid solution and subsequent aging treatment to obtain the antibacterial precipitated phase, which is the key process to endow the stainless steel with antibacterial properties. Since the precipitated phase is uniformly dispersed both on the surface and in the matrix of the material, when wear, abrasion, or machining produces a fresh surface, this in turn enhances the antibacterial activity [62,63,64,65]. However, for antibacterial coatings, including Ag- and Cu-containing coatings, wear and abrasion destroy the coating.

2.Easy control and preparation

The normal processing steps for metals and alloys, including casting, metal forming, and powder metallurgy, can be used to prepare and produce antibacterial metals and alloys with different shapes, including bars, sheets, and even complex structures. The metals and alloys can also be machined to produce food contact devices with complex figurations without a reduction in antibacterial properties. Normal sterilization treatments can also be used for antibacterial metal and alloy devices without a reduction in antibacterial properties.

These advantages have attracted attention to antibacterial alloys as a new kind of food contact material. Thus, many studies on antibacterial alloys have been carried out, both in simulated food and in actual application environments.

### 4.1. Liquid Food Preservation

Tap water, whether used as drinking water or domestic water, is closely related to people’s life and health. The microbial species in tap water are complex and diverse. Common pathogenic bacteria in tap water include *Salmonella typhi*, *Shigella*, *cholerae*, and *Escherichia coli* (*E. coli*) [66]. In recent years, there have been numerous outbreaks of diseases caused by contaminated drinking water. Every day, nearly 4000–6000 people die from diarrhea caused by contaminated tap water; most are children with relatively low immunity [67]. In order to prevent and control the outbreak of waterborne diseases, countries have made large investments in the continuous optimization of the disinfection of tap water. Chlorination is still internationally recognized as a safe, green, and commonly used method. However, the duration and dose of this disinfectant are limited, and it is impossible to comprehensively kill microorganisms without dead ends. Considering that stainless steels occupy an important position in the water treatment and supply pipeline system, Li adopted a Cu-bearing stainless steel (304-Cu SS) in a real tap water environment. Although there are various bacteria in tap water, the 304-Cu SS showed a great inhibitory effect on the total bacteria in the tap water, and thus could maintain self-cleaning by inhibiting the bacterial adhesion on its surface, as shown in Figure 3 [68].

Food is a natural medium for microorganisms. During food production, storage, and sale, the residues of food created due to untimely sterilization or imperfect cleaning provide a beneficial space and resources for bacteria to breed [69,70]. This not only affects the shelf life of food, but also brings in various foodborne pathogenic bacteria [71,72]. Nan et al. proved that 304-Cu SS as a container material could effectively exert its antibacterial properties in the milk environment and maintain the good sanitation of the food’s surrounding environment. Even if *S. aureus* and *Pseudomonas aeruginosa* were added to the milk, more than 99% of the bacteria in the milk medium could be killed, showing that 304-Cu SS is an ideal material for food contact that can reduce food safety risks [73]. Zhao et al. found that after 304 SS and 304-Cu SS with milk drops were placed at 37 °C for 7 days, during the co-culture period, the milk on the surface of the 304 SS presented a phenomenon of solidification and shrinkage. Conversely, the milk on the surface of the 304-Cu SS maintained a stable milky white film, showing a fresher state [74]. In addition, the 304-Cu SS could reduce the adhesion state of residual milk stains on its surface, which might have been related to the lower surface energy of Cu-bearing SS than that of ordinary SS [75]. Similarly to the liquid food media, Zhang et al. placed 430-Cu SS and 430 SS in fresh fruit juice for experiments. As shown in Figure 4, although the colors of all juices were dimmed after co-culturing for 10 days, many bacterial colonies appeared on the surfaces of the juice co-cultured with 430 SS and in the blank control. While there was no visible bacterial colony in the fruit juice co-cultured with 430-Cu SS, there was a slight loss in fruit flavor and a flocculent sediment at the bottom. This result intuitively indicates that 430-Cu SS could significantly delay the spoilage of fruit juice [76].

As for the bactericidal mechanism of antibacterial materials in liquid food media, Zhang et al. studied the application of Cu-bearing SS in a reverse-osmosis (RO) membrane-based water treatment system and confirmed that good antibacterial performance against the planktonic bacteria of the leach liquors was exhibited by 304L-Cu SS, which could alleviate the biological burden from the front end in the liquid environment. The authors also revealed the release-type mechanism of Cu-bearing SS in a liquid system (Figure 5), i.e., the bacteriostasis of Cu-bearing SS resulted from the slow and stable release of Cu ions, and its ionic toxicity and catalytic ROS were key to the excellent bactericidal performance of Cu-bearing SS in liquid media [77].

### 4.2. Solid Food Preservation

Unlike liquid food media, in which antibacterial elements such as Cu ions can diffuse and migrate, whether antibacterial alloys exhibit a preservative function for solid food has also attracted attention. Fresh fruits with various skins types consumed in daily life, including bananas, strawberries, and sugared oranges, were selected by Zhao et al. for simulated preservation experiments to explore whether 304-Cu SS as a food contact material could extend the shelf life of fresh fruits. As shown in Figure 6, after the fresh fruits were placed on two different kinds of food contact plates (304-Cu SS and 304 SS, respectively) and stored at room temperature for different durations, the formation of the banana plaque resulting from *Colletotrichum* spp. could be retarded by the 304-Cu plate, significantly delaying the process of banana rot [78]. Similarly, the fragile strawberry without an outer skin and the sugared orange with a thicker skin, which are commonly believed to rot easily, were also kept fresh for a longer time by the 304-Cu plate [74].

Zhang et al. simulated several typical application scenarios of stainless steel in daily life to explore the advantages of Cu-bearing SS as a food contact material for food preservation. Herein, fresh chicken meats were spread out on antibacterial stainless steel (304-Cu) and ordinary stainless steel (304) sheets, and the changes in the spoilage characteristics of the chicken over time were recorded. It was found that a reduction in the quality of the chicken placed on 304 was significantly greater than that on 304-Cu. For example, the chicken placed on 304 presented a sticky surface with overflowing mucus, giving off a rancid smell; there was an increase in the conductivity of the chicken leaching solution; and the overproduction of total volatile basic nitrogen (TVB-N) by the chicken was observed [79]. Of course, this mucus actually represented a large amount of bacterial colonies formed on the surface of the chicken [80]. The decomposition products, such as ammonia and amine, generated by the spoilage bacteria are usually alkaline nitrogen-containing substances [81]. Additionally, the breaking down of meat proteins and fats induces the generation of large amounts of conductive ions (K^+^, Na^+^, Cl^−^, etc.), resulting in an increase in the conductivity of chicken leaching solution [82], as shown in Figure 7 and Table 1. The above studies fully indicated that Cu-bearing SS as a food contact material could effectively delay the spoilage process of food. The strong and effective inhibition of solid food spoilage resulted from the contact-killing activity of Cu-bearing SS. Numerous studies have claimed that bacteria, yeasts, and viruses can be rapidly killed on antibacterial metallic surfaces [83,84,85]. The main mechanism is contact killing, i.e., the antibacterial surface can effectively inactivate the bacteria through direct and sufficient contact. Dry Cu surfaces have shown high killing efficiency against a wide range of microbes, which is attributed to the faster Cu uptake of bacterial cells on dry Cu than moist Cu [86]. Furthermore, Cu- or Ag-containing nanoscale or microscale particles can form micro-galvanic cells with metal substrates, thus inducing an antibacterial effect by electron transfer. Wang et al. found that the antibacterial effect of Ag-NPs@Ti and the electron transfer taking place between the Ag-NPs and the Ti substrate were the key factors for producing ROS to elevate the oxidative stress among the bacteria [87]. Zhang et al. clarified that Cu-bearing stainless steel could induce electron transfer reactions with bacterial cells, destroying the respiratory chain and generating ROS to hinder bacterial growth, as schematically shown in Figure 8 [84]. Since charge transfer is a short-range interaction, the material can usually effectively inactivate the microorganisms in the food upon contact, thus inhibiting an increase in the total bacteria in the food, contributing to its preservation.

Microbes are indeed an important factor in food spoilage, but only when the microorganisms multiply in the food matrix and accumulate to a certain degree will food spoilage be caused [79]. In essence, food spoilage is a quorum-sensing (QS) phenomenon caused by one or more bacteria. That is, after the accumulation of microorganisms in food to a certain extent, they produce, release, and monitor self-induced signaling molecules to unify and coordinate bacterial population behaviors, so as to express specific spoilage traits such as producing rotten metabolites (odor, pigments, etc.); secreting extracellular enzymes to degrade food nutrients; and forming mucus and biofilms on the surface of food. Studies have shown that when the number of bacterial colonies in food reaches 6lg CFU/mL, the main signal molecule in the QS system of Gram-negative bacteria, N-acylhomoserine lactone (AHL), can be detected [88]. Gram et al. added the *E. carolovora*, which produces AHL and can hydrolyze pectin, to bean sprouts, and the storage test showed that it could accelerate the decay of bean sprouts [89]. They also found that when *Enterobacteriaceae* in meat products reached a certain cell density, AHL molecules could promote the production of an *Enterobacteriaceae* enzyme system, thus contributing to changes in the quality of meat products at a relatively low cell count [90]. Accordingly, the development of novel food preservatives such as specific molecular inhibitors to quench QS signals has become a new research focus in the field of food preservation. Zhang et al. found that a 304-Cu SS container could not only reduce the bacterial colonies grown on bean curd and steamed rice, but also form a bacteriostatic atmosphere around them. In this regard, the effect of 304-Cu SS on an interspecific QS signal (AI-2), which exists widely in foodborne pathogens and spoilage bacteria, was investigated. As shown in Figure 9, it was also found that Cu-bearing SS could quench the bacterial QS signal by producing ROS, which resulted in the bacteria being unable to express spoilage traits and form a biofilm on the contacted food and its surroundings. Thus, the food preservation effect could be enhanced [79].

### 4.3. Actual Industrial Applications

Metal food contact materials are widely used, especially stainless-steel products, which account for more than 90% of food contact materials. They are widely used in tableware, kitchen utensils, food packaging, food production equipment, etc., directly affecting the cleanliness and safety of food [91,92]. Thus, these instruments with antibacterial and self-cleaning behavior are of great significance. In the 1990s, the Kawasaki Iron and Steel Co., Ltd., a Japanese company, first launched Ag-bearing SS. Then, another Japanese company, Nisin Steel Co., Ltd., developed Cu-bearing SS, including an austenitic system (18Cr-9Ni-3.8Cu), ferritic system (17Cr-1.5Cu), and martensitic system (0.3C-13Cr-3Cu). These three groups of stainless steels were named NSSAM series, and they have been confirmed to possess excellent and stable resistance to most common bacteria, such as *E. coli* and *S. aureus* [93,94]. In China, the Institute of Metals Research of the Chinese Academy of Sciences has taken the leading role in developing a series of Cu-bearing antibacterial SSs, including an austenitic system (201-Cu, 301-Cu, 304L-Cu, 316L-Cu, 317L-Cu); ferritic system (430-Cu); martensitic system (2Cr13-Cu, 3Cr13-Cu, 5Cr15-Cu); and even duplex system (2205-Cu). It was also confirmed that the bactericidal rate of Cu-bearing SSs against most microorganisms was over 90%. Furthermore, antibacterial SSs were studied from the perspective of composition design, preparation, microstructure, and properties, which greatly promoted the development of antibacterial SSs [95,96,97]. Moreover, Cu-bearing SSs were also found to have excellent biocompatibility. Sun et al. added 4.5% Cu to 317L austenitic stainless steel, and the 317L-Cu stainless steel was co-cultured with zebrafish after subjection to aging treatment [98]. The results showed that the 317L-Cu had no obvious toxicity to zebrafish. Ren et al. performed in vitro cell compatibility experiment on 317L-Cu SS and found that it could stimulate alkaline phosphatase activity and promote osteogenic gene expression, thus promoting osteoblast differentiation [99]. Therefore, the previous research on antibacterial SSs was mostly focused on their application in the medical field. In 2005, Taiyuan Steel Plant, China, successfully fabricated both ferritic and austenitic antibacterial SSs at the industrial level.

The ferritic antibacterial SS NSS-AM1 showed an excellent antibacterial effect, with the potential to be used in washing machines, food transport, food refrigerators, and commercial kitchens. The martensitic antibacterial SS NSS-AM2 is applicable for manufacturing kitchen knives, scissors, and other household knives. Later, Zhang et al. developed a kind of Cu-bearing ferritic SS that could be used in the food processing industry; kitchens; and the military, especially in catering equipment for field troops [100]. Many evaluations of antibacterial SSs in actual application environments have proved their attractive application performances.

## 5. Chemical Migration of Food Contact Materials

Any chemical components in food contact materials will migrate to the food upon contact based on molecular movement and diffusion. Such migration may affect food safety and food quality and should not be ignored. Antimicrobial packaging in particular allows a release of antimicrobial agents during storage and distribution. This is of the utmost importance, as a rapid release causes the fast consumption of the antimicrobial agents, after which the minimal concentration required for microbial growth inhibition is not maintained on the food surface. On the other hand, spoilage reactions on the food surface may start if the release of the antimicrobial agent from the packaging film is too slow [101]. The special effects of nano-active compounds make them popular in the food field, but attention must be paid to the biological and environmental impacts they may bring. The particle size of nanoparticles is very small, and the small specific surface area greatly enhances their diffusivity and adsorption [102]. For example, the diffusion coefficient of ordinary Cu increases by more than 1000 times when it is made into nano-Cu with a particle size of 8 nm through nanotechnology. Nanomaterials in complex media such as foods and food simulants can be very problematic [103]. However, the legal frameworks for their use in food contact applications are not mature. European Framework Regulation (EC) No. 1935/2004, which lays down the general principles for any material or article intended to come into contact with food, has not yet addressed explicitly the use of nanotechnology related to such food contact materials [104]. In the last few years, many scientific papers studying the potential migration of nanomaterials have been published. In most cases, nanosilver-containing polymers were the object of these studies. However, ZnO, TiO_2_, and other nanoscale additives have also been investigated [105]. The migration of Ag and other elements was quantitatively determined as total silver by element-specific methods (inductively coupled plasma–mass spectrometry (IPC-MS) and atom absorption spectroscopy (AAS)) [106]. Migration analysis is almost mandatory in deciding which synthesized nanofilms are most suitable for which foods, and such suitability analysis should be conducted according to the following methods: (1) 50% ethanol, simulant imitating milk and milk products; (2) 10% ethanol, simulant imitating liquid foods; and (3) 3% acetic acid, simulant imitating acidic foods. The right choice of material is a very effective method to extend the shelf life of foods [107]. Mackevica et al. measured Ag migration from commercial polymer nanocomposites, observing higher migration (up to 0.31 μg/dm^2^) into the acidic medium and non-detectable migration (<0.06 μg/dm^2^) into 10% ethanol and water [108]. Similarly, the migration of nanocomposites has not been shown to be consistent between different synthesized nanofilms. Commercial polypropylene (PP) containers and polyethylene (PE) bags were investigated by von Goetz et al. [109]. For the containers, the authors found a migration of up to 1 μg/dm^2^ in 3% acetic acid (10 days at 20 °C), whereas the migration from the bags was found to just reach the detection limit of 0.05 μg/dm^2^, although the silver content in the PE bags (37 μg/g) was higher than in the PP box (12 μg/g). Based on this, contrary to the generally predicted growth of nano food contact applications, the number of existing and legally authorized materials is rather small. This may have something to do with existing safety concerns about nanoparticles in food contact materials and the fact that there are currently no toxicologically derived thresholds of no concern. Of course, engineering advances are moving toward micro- and nano-encapsulation as innovative ways to obtain a tailored release rate and increase the stability of bioactive compounds [110]. So far, food biopolymers and especially hydrocolloids are promising materials to produce carriers/emitters for antimicrobials [111]. Additionally, numerous polysaccharides and proteins have been mixed with various antimicrobial substances to produce active encapsulations [112]. Nonetheless, specific legislative rules are missing. According to the US Food and Drug Administration (FDA), substances that can come into contact with food (including a variety of packaging materials and containers) are considered safe when the cumulative dietary concentration is less than 0.5 ug/kg [113].

Similarly, for metal food contact materials, the health and safety problem mainly stems from the migration of toxic heavy metal elements [114]. In this regard, the European Union and other countries have mainly stipulated the migration limit of heavy metals, including lead, cadmium, chromium, and nickel, in metal food contact materials. As for migration testing, the Chinese regulations stipulate that no matter what kind of food these products are exposed to, acetic acid is used for migration measurement, in combination with the test conditions of boiling for 30 min and leaving at room temperature 24 h. This was mainly decided by considering that, compared with water, alcohol solutions, and other media, metals are more easily dissolved in acidic solutions. Additionally, the higher the temperature and the longer the exposure duration, the greater the dissolution of metals [115]. As for other elements, such as Ag, Cu, and Zn, which are frequently used in antibacterial alloys, it is only stipulated internationally that the amounts transferred into food should not endanger human health, cause unacceptable changes in food composition, or result in undesirable sensory degradation. In response, safety evaluation data for certain metals have been put forward, such as a provisional maximum tolerable daily intake (PMTDI) and provisional tolerable weekly intake (PTWI), as summarized in Table 2 [116].

In fact, the existence and release of antibacterial agents in antibacterial alloys are relatively stable compared with the behavior of antimicrobials in films or coatings. It has been reported the Cu ion release from any antibacterial titanium alloys is much lower than the recommended daily intake of Cu for an adult shown in Table 2 (3 mg), and the Ag ion concentrations from all Ti-Ag sintered alloys (1–3% Ag) after 24 h immersion were found to be lower than 5 ppb [55]. Meanwhile, the release of Zn ions from Zn alloys is as low as 22.1 ± 4.7 μm/year, which is mainly attributed to the higher standard potential of Zn [117]. Regarding Cu-bearing SS, which has the most promising applications in the food field, Ren et al. determined that the release rate of Cu ions from 304-Cu SS in pure water was only 7.60 μg/L day [118]. In Zhang’s research on the use of Cu-bearing stainless steel as a new strategy to delay food spoilage, the authors reported that the concentration of Cu ions released from 304-Cu SS in phosphate-buffered solution (PBS) was 14 ppb/day, reaching only 32 ppb/day even in a bacterial suspension with a concentration of 10^5^ CFU/mL [89]. They also confirmed the stability of 304L-Cu SS, finding that even if it was immersed for 28 days in corrosive sodium chloride, the release of all metal ions (including Fe, Cr, Ni, and Cu) was at a lower level. In particular, the maximum accumulated concenreation of Cu release was only 69 ppb [77]. In short, the amount of antibacterial metal ions released from antibacterial alloys is far below the maximum safe amount for humans and the body’s bio-safety limits, such as the safe limit of 1300 ppb/day for Cu in drinking water specified by the Environmental Protection Agency (EPA) of USA [119]. Thus, the use of antibacterial alloys developed by existing processes as food contact materials does not constitute a hazard to human health and safety.

## 6. Summary and Prospects

Antibacterial materials have gradually come to play an irreplaceable role in food preservation. However, although the use of antibacterial food contact materials presents multiple advantages, including greatly reducing the risk of food contamination, facilitating food storage and transportation, extending food shelf lives, and even enhancing the original commercial value of food, food contact materials are also known as indirect food additives. The safety of raw materials, auxiliary materials, and processing procedures directly affects the quality of food and human health. Component migration is the key safety issue that must be a priority when developing a new antibacterial food contact material. Compared with antibacterial materials containing natural antibacterial additives, which show limited antibacterial ability, poor stability, and easy migration, metal-based inorganic antibacterial agents have the characteristics of high safety, good heat resistance, a broad antibacterial spectrum, no drug resistance, and no antibacterial failure. In particular, the antibacterial alloys developed in recent years have shown excellent antibacterial durability while ensuring good processing properties and the malleability of the original materials. Antibacterial alloys can realize the comprehensive and effective inhibition of food microorganisms through release-type mechanisms, contact-killing mechanisms, and the regulation of microbial populations. Significantly, antibacterial alloys show extremely low levels of the migration of complementary components such as Ag, Cu, and Zn, which may be attributed to their unique material structure and antibacterial mechanisms. Broad application prospects can be expected, although safety laws and regulations still need to be improved. This will certainly lead to the future development of more efficient, environmentally friendly, and safe functional food contact materials.

## Figures and Tables

**Figure 1 materials-16-03029-f001:**
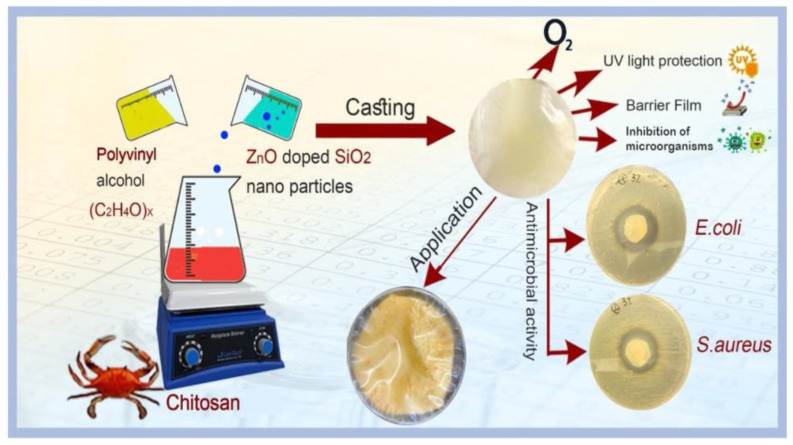
The preparation and utilization of PVA/CS/ZnO-SiO_2_ bio-nanocomposite films for bread preservation [47].

**Figure 2 materials-16-03029-f002:**
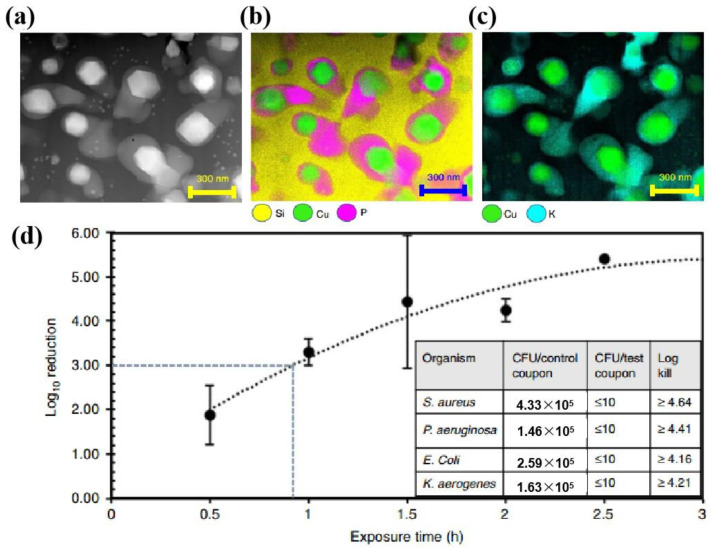
Elemental mapping of chemical species in the microstructure of copper–glass ceramic by energy-dispersive X-ray spectroscopy (EDS) using a scanning transmission electron microscope: (**a**) A microstructural view of the copper–glass ceramic highlighting the three phases present. Some smaller cuprite crystals were also observed in the continuous phase. (**b**) EDS elemental mapping. (**c**) EDS elemental mapping also revealed that the lower-durability phase was enriched in potassium. (**d**) Bacterial reduction kinetics on paint coupons containing copper–glass ceramic particles. Graph shows *S. aureus* reduction as a function of exposure time [54].

**Figure 3 materials-16-03029-f003:**
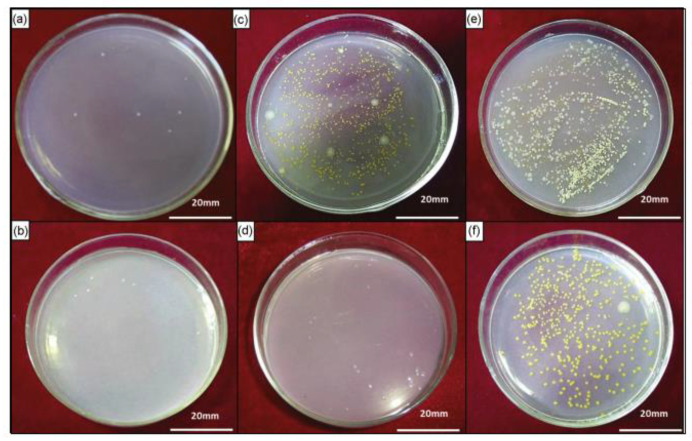
Photos of bacterial colonies on different stainless steels after immersion in the tap water for different durations: (**a**) 304 for 24 h, (**b**) 304-Cu for 24 h; (**c**) 304 for 48 h, (**d**) 304-Cu for 48 h; (**e**) 304 for 72 h, (**f**) 304-Cu for 72 h [68].

**Figure 4 materials-16-03029-f004:**
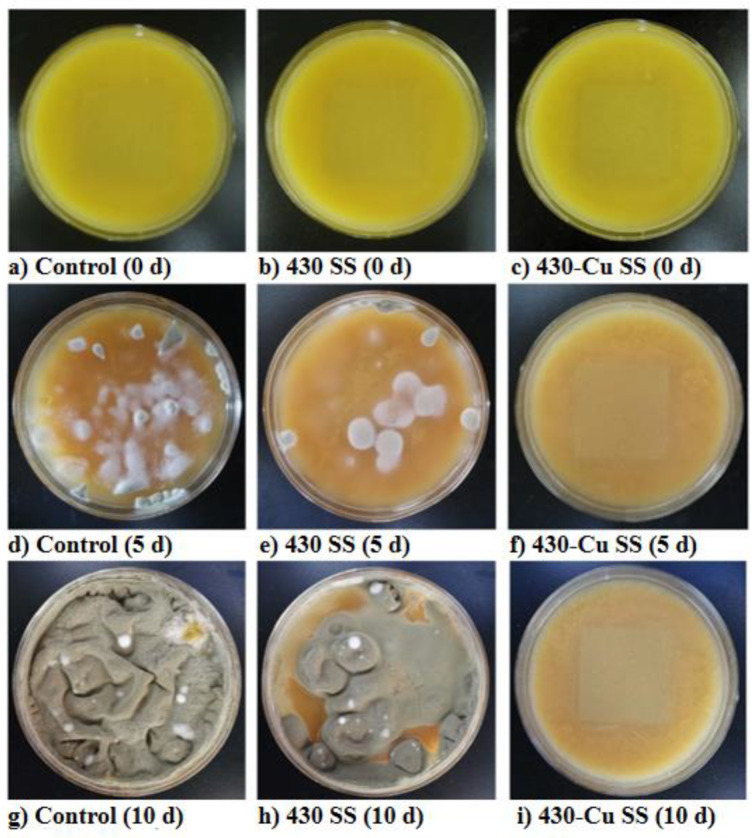
Macroscopic features of orange juice immersed with 430 and 430-Cu stainless steels, respectively, for 5 d and 10 d [76].

**Figure 5 materials-16-03029-f005:**
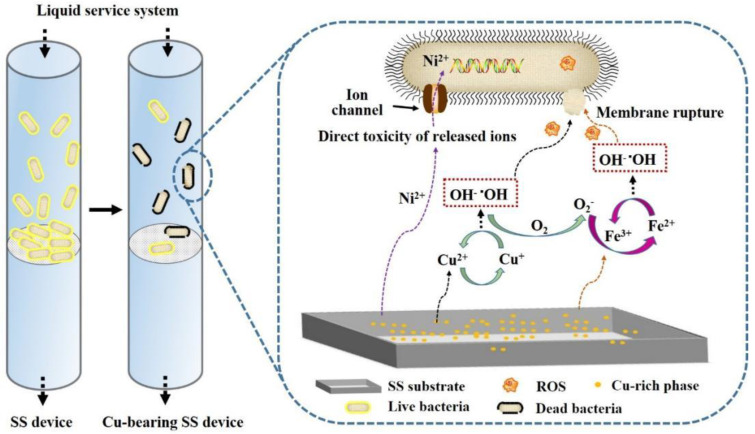
Release-type mechanism of Cu-bearing stainless steel in liquid system [77].

**Figure 6 materials-16-03029-f006:**
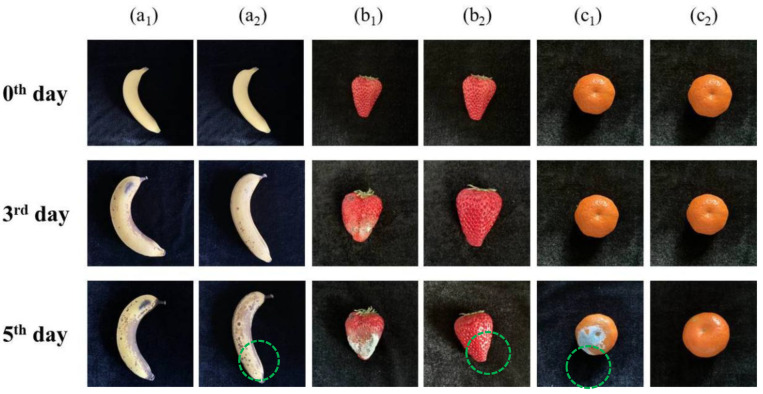
Photos of fruits stored at room temperature over several days: (**a**) banana; (**b**) strawberry; (**c**) sugared orange. Subscripts: (1) on 304 plate; (2) on 304-Cu plate [74].

**Figure 7 materials-16-03029-f007:**
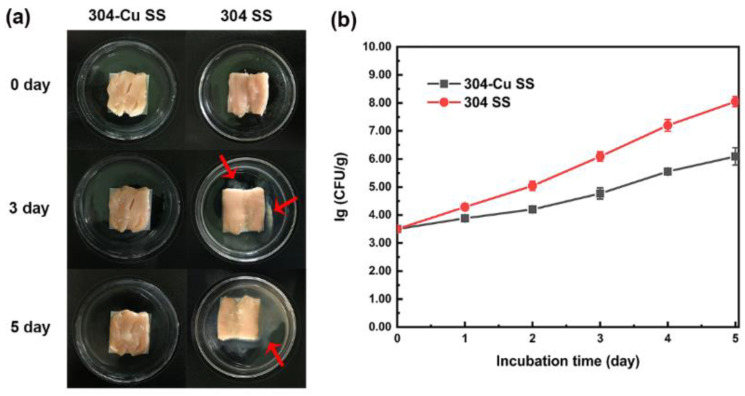
(**a**) Sensory changes in chicken meat spread out on antibacterial 304-Cu and ordinary 304 stainless-steel sheets, respectively; red arrows indicate the areas where the mucus obviously overflowed. (**b**) Changes in total bacterial count of chicken spread out on antibacterial 304-Cu and ordinary 304 stainless-steel sheets [79].

**Figure 8 materials-16-03029-f008:**
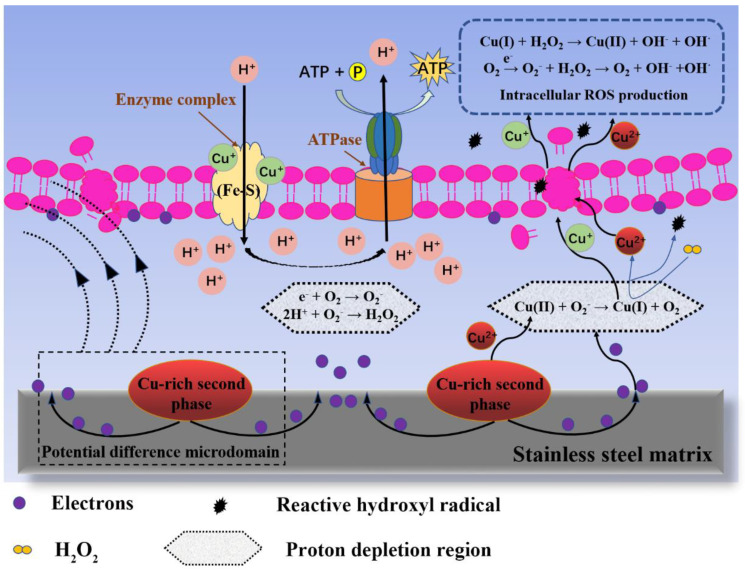
Schematic illustration of the bacteria-killing process of Cu-bearing stainless steel [84].

**Figure 9 materials-16-03029-f009:**
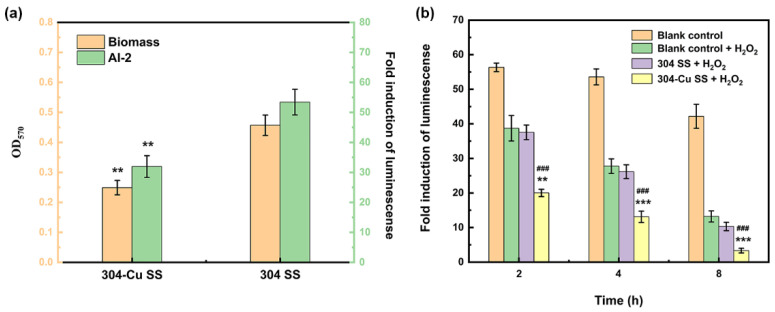
(**a**) Biomass attached to the surfaces of antibacterial 304-Cu and ordinary 304 stainless steels and the corresponding AI-2 signal expression levels. * Indicates significant difference between 304-Cu and 304 (* *p* < 0.05, ** *p* < 0.01, *** *p* < 0.001). (**b**) Changes in AI-2 signal activity under different treatments. All data represent the mean ± standard deviation of three independent experiments. * Indicates significant difference between 304-Cu SS + H_2_O_2_ group and blank control + H_2_O_2_ group (* *p* < 0.05, ** *p* < 0.01, *** *p* < 0.001); # indicates significant difference between 304-Cu SS + H_2_O_2_ group and 304 SS + H_2_O_2_ group (# *p* < 0.05, ## *p* < 0.01, ### *p* < 0.001) [79].

**Table 1 materials-16-03029-t001:** Changes in electrical conductivity and TVB-N of chicken meat spread out on antibacterial 304-Cu and ordinary 304 stainless-steel sheets [79].

Incubation Time (days)	Indicators			
	Electrical Conductivity (μS cm^−1^)		TVB-N (mg 100 g^−1^)	
	304-Cu SS	304 SS	304-Cu SS	304 SS
0	867.6 ± 6.80	867.6 ± 6.80	13.22 ± 0.43	13.22 ± 0.43
1	947.6 ± 19.19	1090.4 ± 3.62	13.87 ± 0.11	14.76 ± 0.22
2	1354.2 ± 21.34	1567.2 ± 36.94	14.17 ± 0.26	16.31 ± 0.57
3	1457.3 ± 15.84	1771.8 ± 27.54	17.64 ± 0.71	21.49 ± 2.12
4	1608.9 ± 13.85	1990.6 ± 16.11	20.33 ± 0.68	30.48 ± 0.97
5	2148.0 ± 16.43	3312.0 ± 14.83	25.49 ± 1.48	42.66 ± 1.32

**Table 2 materials-16-03029-t002:** Safety evaluation data for specific metals and alloys [116].

Element	Daily Average Value (mg/d)	Weekly Average Value (mg/week)	PMIDI (mg/(kg·bw·d))	PIWI (mg/(kg·bw·week))
Al	6	42	-	7
C	0.2	1.4	-	-
Cu	3	21	0.5	-
Fe	15	105	0.8	-
Pb	0.05	0.35	-	0.025
Mn	-	-	-	-
Ni	0.4	2.8	0.005	-
Ag	0.007	0.05	-	-
Sn	4	28	-	14
Ti	0.8	5.6	-	-
Zn	17	119	1	-
Be	-	-	-	-
Co	1	7	-	-
Cd	0.015	0.105	-	0.007
Hg	0.01	0.07	-	0.005

## Data Availability

The data presented in this study are available on request from the corresponding author.

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
