# Peer review of "Novel Antibacterial Metals as Food Contact Materials: A Review"

_materials, 2023, doi:10.3390/ma16083029_

Round 1
Reviewer 1 Report
Comments to Author:
I would suggest to add a subsection on actual industrial applications of food contact materials. Some examples are already given in subsection 2 Easy control and preparation. These descriptions of the companies that produce the aforementioned food contact matarials and their products, as well as an additional review of applications in this area, should be summarized in a special subsection.
Summary and prospects
The summery should be more impressive. Emphasize the difference and uniqueness of your review article on the topic of new antibacterial metals as a food contact materials in comparison with other review articles on same topic. What makes your review article special, what does it highlight differently than others.
Reviewer 2 Report
This manuscript written by Xinrui Zhang et al. is a research on “Novel Antibacterial Metals as Food Contact Materials: A Review”. I believe that it will be very helpful the major revision. comments are listed below
1. Unfortunately, there is very little research in the last 5 years in the introduction section, which causes the research to stay away from new methods. Please most of your references needs to renew.
2. Ag, Cu and Zn are frequently used as antibacterial agents, especially in bionanofilms synthesized to extend the shelf life of foods. However, the use concentrations of these agents are of vital importance. Attention should be paid to the fact that the agents used are in concentrations that will not harm the content of the food and health, and a detailed explanation should be given about the optimum concentrations.
3. Similarly, detailed explanation is needed for antibacterial alloys.
4. In recent years, especially encapsulation or microencapsulation techniques have emerged as innovative methods to extend the shelf life of foods. Unfortunately, there is not enough explanation in this sense in the manuscript. Please clarify this issue by referring to more recent literature.
5. Another major deficiency in the manuscript is migration analysis, which is the most important parameter in food packaging. As it is known, migration analysis is almost mandatory in deciding which bionanofilms synthesized are more suitable for which foods. Especially in migration analysis, analysis suitability should be measured according to the following items. This right choice is a very effective method to extend the shelf life of foods.
50% ethanol; simulant imitating milk and milk products,
10% ethanol; simulant imitating liquid foods,
3% acetic acid; simulant imitating acidic foods
It would be helpful to use the following research
Proporties and Synthesis of Biosilver Nanofilms for Antimicrobial Food Packaging
G Baysal, C Demirci, H Ozpinar
Polymers 15 (3), 689
6. In the abstract section, the migration features need to be explained.
7. In general, plagiarism isn’t high in your manuscript, nevertheless It would be helpful to more improve
8. Minor grammatical errors in English need to be corrected. please check well

Reviewer 3 Report
The manuscript deals with a review regarding novel antibacterial metals as food contact materials.
The manuscript is interesting. The biocompatibility of novel antibacterial metals as food contact materials should be presented in more detail and their advantages and drawbacks must be highlighted. The authors must also point out better new directions for the use of these antibacterial metals to improve food packaging systems.
The English language must be revised.
Reviewer 4 Report
Novel Antibacterial Metals as Food Contact Materials.
The manuscript is a review but the data concerns only the area of ​​China or it should be included in the title or the data should be extended to the whole world.
The manuscript is valuable but requires a lot of improvement. I can recommend it after major revision.
Incorrect chemical and biological notations.
A lot of old literature. Literature of only authors from one country predominates, it is not very professional. Science knows no boundaries.
it is worth giving newer items. For example:
Effect of chitosan, hyaluronic acid and/or titanium dioxide on the physicochemical characteristic of phospholipid film/glass surface, Physicochem. Probl. Miner. Process. 55(6) (2019) 1535–1548
Release kinetics and antimicrobial properties of the potassium sorbate-loaded edible films made from pullulan, gelatin and their blends, Food Hydrocolloids 101(2020) 105539 140
Pay special attention to the drawings and diagrams in the review work, this is the most important thing, this is what potential readers pay attention.
I can recommend manuscript after major revision.
Round 2
Reviewer 2 Report
Dear Authors, I am pleased to inform accept of revised manuscript.
Reviewer 3 Report
The manuscript was improved.
Reviewer 4 Report
References,There are errors in the names of the authors, e.g 36, 101,...